# Genetic Variation of Native *Perilla* Germplasms Collected from South Korea Using Simple Sequence Repeat (SSR) Markers and Morphological Characteristics

**DOI:** 10.3390/plants10091764

**Published:** 2021-08-25

**Authors:** Jun Seok Oh, Kyu Jin Sa, Hyeon Park, Do Yoon Hyun, Sookyeong Lee, Ju Hee Rhee, Ju Kyong Lee

**Affiliations:** 1Department of Applied Plant Sciences, College of Agriculture and Life Sciences, Kangwon National University, Chuncheon 24341, Korea; dhwnstjr0427@naver.com (J.S.O.); sakajin49@kangwon.ac.kr (K.J.S.); znfnfn325@naver.com (H.P.); 2Interdisciplinary Program in Smart Agriculture, Kangwon National University, Chuncheon 24341, Korea; 3National Agrobiodiversity Center, National Institute of Agricultural Sciences, RDA, Jeonju 54874, Korea; dyhyun@korea.kr (D.Y.H.); xsanta7@korea.kr (S.L.); rheehk@korea.kr (J.H.R.)

**Keywords:** *Perilla* crop, genetic resources, morphological traits, principal component analysis, SSR marker, genetic variation

## Abstract

Using morphological characteristics and simple sequence repeat (SSR) markers, we evaluated the morphological variation and genetic diversity of 200 *Perilla* accessions collected from the five regions of South Korea and another region. In morphological characteristics analysis, particularly leaf color, stem color, degree of pubescence, and leaf size have been found to help distinguish the morphological features of native *Perilla* accessions cultivated in South Korea. Twenty SSR primer sets confirmed a total of 137 alleles in the 200 *Perilla* accessions. The number of alleles per locus ranged from 3 to 13, with an average number of alleles per locus of 6.85. The average genetic diversity (GD) was 0.649, with a range of 0.290–0.828. From analysis of SSR markers, accessions from the Jeolla-do and Gyeongsang-do regions showed comparatively high genetic diversity values compared with those from other regions in South Korea. In the unweighted pair group method with arithmetic mean (UPGMA) analysis, the 200 *Perilla* accessions were found to cluster into three main groups and an outgroup with 42% genetic similarity, and did not show a clear geographic structure from the five regions of South Korea. Therefore, it is believed that landrace *Perilla* seeds are frequently exchanged by farmers through various routes between the five regions of South Korea. The results of this study are expected to provide interesting information on the conservation of these genetic resources and selection of useful resources for the development of varieties for seeds and leafy vegetables of cultivated *Perilla frutescens* var. *frutescens* in South Korea.

## 1. Introduction

*Perilla frutescens* (L.) Britt. is a self-fertilizing species of the Labiate family. *Perilla* species has been traditionally cultivated and used, mainly in Asian countries, especially in East Asia, and is divided into two cultivated types of *Perilla frutescens* var. *frutescens* and var. *crispa*, based on their distinct morphological features [1,2,3,4]. For example, *Perilla frutescens* var. *frutescens* is used as a leafy vegetable and oil crop, and *Perilla*
*frutescens* var. *crispa* is used as a vegetable and Chinese (or herbal) medicine [5,6,7]. These two cultivated types of *Perilla* crop have a very long cultivation history in East Asia [2,6,7,8]. Although the *Perilla* species is widely distributed and used in the Himalayan hills, East Asia, and Southeast Asia, the *Perilla* crop is considered to originate in East Asia because many genetic resources of the two cultivated and weedy types of *Perilla* crop are distributed and used in the region as a traditional folk crop (oil crop, herbal medicine crop, or leafy vegetable crop) [2,3,4,5,6,7]. Furthermore, the two cultivated types of *Perilla* crop are different for their morphological characters. Cultivated *Perilla frutescens* var. *frutescens* has green leaves and stem, non-wrinkled leaves, and a fragrance specific to the var. *frutescens* (Appendix A). In contrast, cultivated *Perilla frutescens* var. *crispa* has red or green leaves, wrinkly or non-wrinkly leaves, and a fragrance specific to the var. *crispa* (Appendix A). Even though the two cultivated types of *Perilla* crop have different morphological features, they are both tetraploid with the same number of chromosomes (2n = 40) [2,4,6].

Today, the two cultivated and weedy types of *Perilla* crop are widely distributed in East Asia, and they are mainly cultivated and used in South Korea and Japan [3,5,6,9]. That is, *Perilla*
*frutescens* var. *frutescens* is widely cultivated in South Korea as both a seed oil crop and a leaf vegetable crop. The *Perilla* leaves of *Perilla frutescens* var. *frutescens* are preferred as a salad vegetable to eat with meat or sashimi mainly in South Korea. Furthermore, *Perilla* seed oil of *Perilla frutescens* var. *frutescens* has a high content of linolenic acid, which is a polyunsaturated fatty acid such as linoleic acid (18:2) and α-linolenic acid (18:3) and comprises approximately 80% of *Perilla* seed oil [10,11,12,13]. Additionally, *Perilla* seeds are used like sesame seeds for flavoring agents or seasoning in Korean traditional foods. Therefore, the cultivation area of *Perilla*
*frutescens* var. *frutescens* has increased greatly, and its leaves and seed oil have been in the limelight as health foods in South Korea [9,13,14]. On the other hand, *Perilla frutescens* var. *crispa* is widely cultivated in Japan and used for leafy vegetables or pickles [4,6].

The conservation of genetic resources is primarily aimed at ensuring the proper preservation and storage of genetic resources belonging to economically important crops that have been collected in genebanks [14,15,16]. In addition, for the successful use and selection of genetic resources that can be used for crop improvement, it is most important to evaluate and understand the genetic diversity (GD) of genetic resources preserved in genebanks [15,16]. In *Perilla* crops, many researchers have analyzed GD, genetic relationships, population structure, and association mapping analysis of accessions of the two cultivated and weedy types of *Perilla* crop using several DNA molecular marker systems including random amplification of polymorphic DNAs (RAPD) [17], amplified fragment length polymorphisms (AFLP) [3], and simple sequence repeats (SSRs) [9,13,14,18,19,20]. Among these DNA molecular markers, particularly SSR (or microsatellite) markers have provided useful information for the analysis of GD, genetic relationships, population structure, and association mapping analysis in the accessions of the two cultivated types of *Perilla* crop and their weedy types, because *Perilla* SSR markers have been shown to be reproducible, highly polymorphic, and generally codominant in *Perilla* accessions [19,20,21,22,23]. Therefore, *Perilla* SSR primers are expected to be useful molecular markers for the analysis of genetic variation (or diversity), genetic relationships, and population structure of the germplasm of *Perilla frutescens* var. *frutescens* of *Perilla* crop.

Furthermore, to preserve and utilize efficiently the genetic resources of *Perilla* crop stored in the RDA-Genebank of the Republic of Korea (http://genebank.rda.go.kr/, accessed on 1 December 2019), it is necessary to understand the morphological and genetic variations among native accessions of *Perilla frutescens* var. *frutescens* in South Korea. In particular, morphological variations in cultivated crop species and their cultivars or native varieties can be affected by the growing environment, geographic distribution, and cultivation history within their cultivated area or habitat [24,25,26,27]. Thus, it is necessary to clarify the geographical patterns of morphological variations of germplasm resources of cultivated *P. frutescens* var. *frutescens* in South Korea for the efficient management and selection of these germplasms.

Therefore, in this study, we used morphological characteristics and *Perilla* SSR markers to evaluate the morphological variation and GD according to the geographical distribution of germplasm resources of native *Perilla frutescens* var. *frutescens* of *Perilla* crop collected in South Korea. The results of this study are expected to provide useful information for understanding the genetic variation of native *Perilla* germplasm collected from South Korea.

## 2. Results

### 2.1. Morphological Variation in the Accessions of Perilla Frutescens Var. Frutescens of Perilla Crop

The results of examining eight qualitative and three quantitative traits for 183 accessions of native *Perilla frutescens* var. *frutescens* are summarized in Appendix A.

In the survey of eight qualitative traits, for color of leaf surface (QL1), 22 *Perilla* accessions showed a light green color, 100 *Perilla* accessions showed a green color, and 61 *Perilla* accessions showed a dark green color. For color of reverse side leaf (QL2), 65 *Perilla* accessions showed a light green, 103 *Perilla* accessions showed green, and 15 *Perilla* accessions showed a dark green. For stem color (QL3), 22 *Perilla* accessions showed a light green, 100 *Perilla* accessions showed green, and 61 *Perilla* accessions showed a dark green. For seed color (QL4), 17 *Perilla* accessions showed white, 8 *Perilla* accessions showed gray, 115 *Perilla* accessions showed brown, and 43 *Perilla* accessions showed a dark brown. For leaf shape (QL5), 97 *Perilla* accessions showed a lanceolate shape, 70 *Perilla* accessions showed a heart shape, and l6 *Perilla* accessions showed an oval shape. For degree of pubescence (QL6), 72 *Perilla* accessions showed slightly pubescent, 108 *Perilla* accessions showed normal pubescent, and three *Perilla* accessions showed heavily pubescent. For seed hardness (QL7), 174 *Perilla* accessions had soft seeds and nine *Perilla* accessions had hard seeds. In the case of plant fragrance (QL8), 167 *Perilla* accessions showed the plant fragrance of *Perilla frutescens* var. *frutescens* and 16 *Perilla* accessions showed a slightly different plant fragrance (Appendix A).

In the survey of three quantitative traits, leaf width (QN1) of the 183 *Perilla* accessions was distributed as follows: 50 *Perilla* accessions were less than 8.0 cm, 69 *Perilla* accessions were 8.1–9.0 cm, 43 *Perilla* accessions were 9.1–10.0 cm, 13 *Perilla* accessions were 10.1–11.0 cm, six *Perilla* accessions were 11.1–12.0 cm, and six *Perilla* accessions were 12.1–13.0 cm. For leaf length (QN2), the 183 *Perilla* accessions were distributed as follows: six *Perilla* accessions were less than 10.0 cm, 23 *Perilla* accessions were 10.1~11.0 cm, 28 *Perilla* accessions were 11.1~12.0 cm, 53 *Perilla* accessions were 12.1~13.0 cm, 42 *Perilla* accessions were 13.1~14.0 cm, 19 *Perilla* accessions were 14.1~15.0 cm, and 12 *Perilla* accessions were more than 15.1 cm. For flowering time (QN3), the 183 *Perilla* accessions were divided into two types as follows: 104 *Perilla* accessions showed intermediate maturing type (flowering days from 15 August to 5 September) and 79 *Perilla* accessions showed late maturing type (flowering days after 6 September). We did not find any early maturing type among the 183 *Perilla* accessions used in this study (Appendix A).

### 2.2. Cluster and Correlation Analysis among 183 Accessions of Native Perilla Frutescens Var. Frutescens Using Morphological Characteristics

In order to understand the morphological difference in accordance with geographic distribution for native Korean *Perilla* accessions, cluster analysis was performed on the 183 accessions of native *Perilla frutescens* var. *frutescens* from five regions of South Korea and another region using 11 morphological traits. The 183 accessions of native *Perilla frutescens* var. *frutescens* clustered into five main groups (Figure 1). Among the five main groups, Group I, containing 143 *Perilla* accessions, was further subdivided into three subclusters. The first subcluster (G I-1) included 88 *Perilla* accessions from the five regions of South Korea and another region. The second subcluster (G I-2) included 48 *Perilla* accessions from five regions of South Korea and another region. The third subcluster (G I-3) included seven *Perilla* accessions from three regions of South Korea. Group II contained six *Perilla* accessions from three regions of South Korea and another region. Group III contained 18 *Perilla* accessions from four regions of South Korea and another region. Group IV contained 12 *Perilla* accessions from three regions of South Korea and another region. Group V contained four *Perilla* accessions from three regions of South Korea (Figure 1).

According to our results, the clustering patterns observed in this study were not clearly distinguished among the native accessions of *Perilla frutescens* var. *frutescens* from the five regions of South Korea and another region. Additionally, most *Perilla* accessions included in each group did not show a specific morphological difference between groups.

Meanwhile, correlation analysis was performed to evaluate the relationships among the 11 qualitative and quantitative traits in the 183 accessions of native *Perilla frutescens* var. *frutescens.* Among all combinations, the combinations of QL1 and QL3 (1.00 **), OL4 and OL5 (0.191 **), OL4 and QN1 (0.239 **), QL4 and ON2 (0.146 *), QL6 and QL7 (0.155 *), QL6 and QN3 (0.166 *), QN1 and QN2 (0.534 **), QN1 and QN3 (0.172 *), and QN2 and QN3 (0.162 *) showed a higher or lower positive correlation coefficient than the other combinations, with a significance level of 0.01 or 0.05. On the other hand, only one combination, QL5 and QN3 (−0.153 *) showed a lower negative correlation coefficient compared with the other combinations. In summary, most of the morphological traits used in the analysis, with the exception of leaf and stem color (QL1, QL3), showed a very low correlation in the native Korean *Perilla* germplasm (Table 1).

### 2.3. SSR Variation in Accessions of Native Perilla Frutescens Var. Frutescens of Perilla Crop

In total, 20 *Perilla* SSR loci were used to evaluate the genetic variation among the 200 accessions of native *Perilla frutescens* var. *frutescens* (Table 2). The 20 *Perilla* SSR primer sets confirmed that a total of 137 alleles were detected segregating in the 200 accessions of native *Perilla frutescens* var. *frutescens*. The number of alleles per locus varied widely from 3 (KNUPF26, KNUPF77) to 13 (KNUPF10) with an average number of alleles per locus of 6.85 (Table 2). The average major allele frequency (MAF) was 0.470, ranging from 0.225 to 0.835. The GD value for each locus ranged from 0.290 to 0.828 with an average of 0.649. The polymorphic information content (PIC) value for each locus ranged from 0.272 to 0.805 with an average of 0.599 (Table 2). Among the 137 alleles, 31 private alleles (23%) were only detected in one of the 200 accessions of native *Perilla frutescens* var. *frutescens*. The percentage of rare alleles (frequency < 0.05) was 45% (62 alleles) among the 137 alleles. On the other hand, the intermediate-frequency alleles (frequency of 0.05–0.5) and abundant alleles (frequency > 0.5) represented 50% (68 alleles) and 5% (7 alleles), respectively, of the total alleles (Appendix A).

To compare the geographical genetic variation of native *Perilla frutescens* var. *frutescens* collected from the five regions of South Korea and other region, this study also confirmed the number of alleles, MAF, GD, and PIC among the 200 accessions of native *Perilla frutescens* var. *frutescens* (Table 3). The average number of alleles was four for the native accessions of *Perilla frutescens* var. *frutescens* from the G1, G2, and G3 regions of South Korea and five for the G4 and G5 regions of South Korea and theG6 region. The average MAF values were 0.537 for accessions of the G1 region, 0.508 for accessions of the G2 region, 0.555 for accessions of the G3 region, 0.497 for accessions of the G4 region, 0.500 for accessions of the G5 region, and 0.601 for accessions of the G6 region. The average GD values were 0.589 for accessions of the G1 region, 0.594 for accessions of the G2 region, 0.561 for accessions of the G3 region, 0.623 for accessions of the G4 region, 0.620 for accessions of the G5 region, and 0.544 for accessions of the G6 region. The average PIC values were 0.538 for accessions of the G1 region, 0.530 for accessions of the G2 region, 0.505 for accessions of the G3 region, 0.570 for accessions of the G4 region, 0.565 for accessions of the G5 region, and 0.497 for accessions of the G6 region (Table 3). From the results from among the six regions, the accessions of the G4 and G5 regions showed a relatively high GD value compared with those from the other regions of South Korea and the other region, while the accessions of the G6 region showed the lowest GD value.

### 2.4. Population Structure and Genetic Relationships among Accessions of Native Perilla Frutescens Var. Frutescens from Five Regions of South Korea and Other Region

In the population structure among the 200 accessions of native *Perilla frutescens* var. *frutescens* from the five regions of South Korea and another region, it was found that the highest value of *ΔK* for the 200 accessions of native *Perilla frutescens* var. *frutescens* was for *K* = 3 (Figure 2). From the results, all *Perilla* accessions were divided into three main groups and an admixed group at *K* = 3 as follows: Group I included 47 *Perilla* accessions from the G1 (one accession), G2 (seven accessions), G3 (two accessions), G4 (14 accessions), and G5 (22 accessions) regions of South Korea and the G6 (one accession) region. Group II comprised 37 *Perilla* accessions from the G3 (one accession) and G5 (two accessions) regions of South Korea and the G6 (34 accessions) region. Group III included 86 *Perilla* accessions from the G1 (16 accessions), G2 (four accessions), G3 (14 accessions), G4 (19 accessions), and G5 (31 accessions) regions of South Korea and the G6 (two accessions) region. The admixed group included a total of 28 *Perilla* accessions, which came from the G2 (two accessions), G3 (three accessions), G4 (three accessions), and G5 (12 accessions) regions of South Korea and the G6 (eight accessions) other region (Figure 3).

On the other hand, in the dendrogram created using UPGMA, the 200 accessions of native *Perilla frutescens* var. *frutescens* were found to cluster into three main groups and an outgroup with 42% genetic similarity (Figure 4). For the three main groups, Group I included 145 *Perilla* accessions from five regions of South Korea and another region; Group II included 19 *Perilla* accessions, which consisted of one accession from the G1 region, two accessions from the G2 region, eight accessions from the G4 region, and eight accessions from the G5 region; and Group III included four *Perilla* accessions, which consisted of two accessions from the G3 region and two accessions from the G4 region. The outgroup included 32 *Perilla* accessions, which consisted of one accession from the G3 region, four accessions from the G4 region, seven accessions from the G5 region, and 20 accessions from the G6 region (Figure 4).

Furthermore, the *Perilla* accessions (145) of Group I were further subdivided into three groups with a genetic similarity of 45% as follows: G I-1 included 82 *Perilla* accessions and consisted of eight accession from the G1 region, five accessions from the G2 region, 11 accessions from the G3 region, seven accessions from the G4 region, 27 accessions from the G5 region, and 24 accession from the G6 region. G I-2 included 52 *Perilla* accessions and consisted of six accession from the G1 region, six accessions from the G2 region, five accessions from the G3 region, 14 accessions from the G4 region, and 21 accessions from the G5 region. G I-3 included 11 *Perilla* accessions and consisted of two accession from the G1 region, two accessions from the G3 region, five accessions from the G5 region, one accession from the G4 region, and one accession from the G5 region. The outgroup contained 32 *Perilla* accessions and divided into many subgroups as follows: three regions (one accession for G1, four accessions for G4, seven accessions for G5) of South Korea and another region (20 accessions for G6). As the results show, although some of the *Perilla* accessions collected in South Korea were included in the same group in accordance with geographic distribution, most *Perilla* accessions tended to be grouped differently from their collection areas. Meanwhile, the 45 *Perilla* accessions collected from the other region were divided into two groups in the phylogenetic tree (Figure 4). That is, some *Perilla* accessions belonged to the same group as the *Perilla* accessions collected in the five regions of South Korea, and the rest of the *Perilla* accessions were included in the outgroup along with some *Perilla* accessions collected in the five regions of South Korea. Therefore, the results of the clustering patterns revealed in this study were not clearly distinguished between and within the native accessions of *Perilla frutescens* var. *frutescens* from the five regions of South Korea and the other region.

### 2.5. Correlation between Perilla SSR Markers and Perilla Morphological Traits

The correspondence between the *Perilla* morphological traits and *Perilla* SSR-based similarity coefficient matrices was tested in a correlation analysis. The correlation coefficient was 0.01 (Figure 5). In our study, the *Perilla* SSR marker-based clustering of accessions of native *Perilla frutescens* var. *frutescens* shows no similarity to the dendrogram topologies based on the morphological index, although there were some commonalities in the positioning of some *Perilla* accessions in the main group.

## 3. Discussion

### 3.1. Morphological Variation of Native Accessions of Perilla Frutescens Var. Frutescens in South Korea

Morphological variation of many crop species within geographic distribution and cultivation areas is of interest in the study of crop species evolution and differentiation [25,26,27,28]. In particular, domestication is a co-evolutionary process that occurs when wild species are brought into cultivation by humans. This process occurs when wild species are exposed to new selective environments related to human cultivation and use that are essential for human survival [29,30]. Thus, cultivated crops from wild species are a direct product of human choice made over a long time [24,26,27]. This domestication process by humans for wild species has produced cultivated crops with unique features that make them suitable for agriculture [30,31]. Therefore, the domestication of crops is a special case of crop evolution in which crop species adapt to human control and breed in human-controlled environments to support human survival.

As explained in the Introduction, in South Korea the leaves and seed oil of cultivated types of *Perilla frutescens* var. *frutescens* have been in the limelight as health foods because of the increased consumption of meat and the development of various cooking and usage methods for *Perilla* fresh leaves and seeds [9,13,14]. Therefore, the RDA-Genebank of South Korea is building a core population of genetic resources that can develop leaf and seed varieties among the preserved *Perilla* germplasm accessions. For the study of morphological variation of cultivated *Perilla frutescens* var. *frutescens*, we surveyed the morphological characteristics among 200 accessions of native *Perilla frutescens* var. *frutescens* that were preserved in RDA-Genebank and collected from South Korea.

In this study, 183 accessions of native *Perilla frutescens* var. *frutescens* were examined for eight qualitative and three quantitative characteristics. It was found that most native accessions of *Perilla frutescens* var. *frutescens* were morphologically different in most of the morphological traits studied (Appendix A). Among the morphological traits investigated in our study, the native accessions of *Perilla frutescens* var. *frutescens* showed high frequency in the following traits: green for leaf and stem color, brown and dark brown for seed color, lanceolate and heart for leaf shape, slightly pubescent and normal pubescent for degree of pubescence, soft seed for seed hardness, a specific plant fragrance of *Perilla frutescens* var. *frutescens* for plant fragrance, a size of 8.1–10.0 cm for leaf width, a size of 12.1–14.0 cm for leaf length, and intermediate maturing type for flowering time. According to our results, most native accessions of *Perilla frutescens* var. *frutescens* showed slight differences in the eight qualitative and three quantitative traits (Appendix A).

These results indicate that the native accessions of *Perilla frutescens* var. *frutescens* exhibiting highest frequency characteristics for each morphological trait are related to their use as leaves or seeds in South Korea. For example, in South Korea, native *Perilla frutescens* var. *frutescens* is used similarly to vegetables such as lettuce; therefore, it is thought that many *Perilla* accessions showing green leaves and stems are favored and cultivated by farmers. In particular, in the correlation analysis of 11 morphological traits, QL1 and QL3 (1.000 **) related to leaf and stem color showed a very high correlation compared with the other traits. As shown in Figure 1, the clustering patterns observed in this study were not clearly distinguished among the native accessions of *Perilla frutescens* var. *frutescens* from the five regions of South Korea and another region. Therefore, the morphological characteristics of native *Perilla*
*frutescens* var. *frutescens* used in South Korea are similarly favored by farmers in all regions. In addition, the seeds of *Perilla frutescens* var. *frutescens* are used as oil or seasoning like sesame seeds; therefore, it is thought that many *Perilla* accessions with seeds showing soft and brown or dark brown colors are favored and cultivated by farmers. Lee et al. (1991) [10] reported that accessions of *Perilla frutescens* var. *frutescens* with brown or dark brown seeds were known to have a higher seed oil yield than those with white or gray seeds, while those with white seeds were known to have a higher protein content than those with black seeds. Furthermore, according to a report of field surveys by Lee et al. (2007) [32], accessions of *Perilla frutescens* var. *frutescens* showing brown and dark brown seed types were found more frequently than those showing white or gray seed types, and also they were widely distributed from northern to southern areas of South Korea. Meanwhile, white seed type accessions were found mainly in the southern part of South Korea. In interviews with farmers in the southern part of South Korea, *Perilla* seeds were found to be mainly used for seasoning and sesame seeds were used for cooking oil [32]. Therefore, in order to use seed oil or leafy vegetable varieties in South Korea, farmers are thought to have selected native accessions of *Perilla frutescens* var. *frutescens* in accordance with their use in each region of South Korea.

### 3.2. Genetic Variation and Population Structure of Cultivated P. Frutescens Var. frutescens

Information on the genetic variation and population structure of native accessions of *Perilla frutescens* var. *frutescens* is an important basis for improvement of *Perilla* crops in South Korea. In particular, to maximize the use of genetic resources preserved in the RDA-Genebank of South Korea in breeding research, the genetic variation and population structure among native accessions of *Perilla frutescens* var. *frutescens* must be studied. In this study, we used 20 *Perilla* SSR markers for GD, genetic relationships, and population structure among native accessions of *Perilla frutescens* var. *frutescens*. As explained in the Introduction, SSR marker technology has many advantages over the other molecular marker systems: simple experimental methods, high reproducibility, polymorphic genetic information contents, the codominant nature of SSR polymorphisms, and their abundance and distribution in plant genomes [19,21,22]. These advantages make them a suitable tool for studying GD, genetic relationships, population structure, bulk segregant analysis, and association mapping analysis in *Perilla* crops [9,14,20,23,33,34]. In our study, we found a total of 137 alleles with 20 *Perilla* SSRs isolating in the 200 accessions of native *Perilla frutescens* var. *frutescens* examined, with an average of 6.85 alleles per locus. This value of alleles of the 20 *Perilla* SSRs detected is slightly higher than the number of valid alleles per SSR locus found in other crops, such as the values of 3.6 in barley [35], 6.7 in wheat [36], 4.8 in soybean [37], and 4.8 in pigeonpea [38]. As a result, it is thought that the native accessions of cultivated *P. frutescens* var. *frutescens* in South Korea have maintained relatively high genetic variation, as well as showing high variations in morphological characteristics.

In our study, we analyzed the number of alleles, GD, PIC, and MAF among 200 accessions of native *Perilla frutescens* var. *frutescens* collected from the G1 region (19 accessions), G2 region (13 accessions), G3 region (20 accessions), G4 region (36 accessions), and G5 region (67 accessions) of South Korea and G6 region (45 accessions) to compare the geographical genetic variation of native accessions of *Perilla frutescens* var. *frutescens* among the five regions of South Korea and another region (Table 3). The average GD values were 0.589, 0.594, 0.561, 0.623, 0.620, and 0.544, respectively, for the native accessions of *Perilla frutescens* var. *frutescens* from the five regions of South Korea and the other region. Additionally, the average PIC values were 0.538, 0.530, 0.505, 0.570, 0.565, and 0.497, respectively, for the native accessions of *Perilla frutescens* var. *frutescens* from the five regions of South Korea and the other region. Although there were differences in the number of native *Perilla* accessions collected and analyzed in the five regions of South Korea and another region, the *Perilla* accessions of the G4 region showed the highest GD value among the six regions, while the *Perilla* accessions of the G6 region showed the lowest GD value. In addition, among the six regions, in particular the *Perilla* accessions collected in the southern regions of Jeolla-do (G4) and Gyeongsang-do (G5) showed higher GD compared with the other regions (Table 3). Therefore, the present evaluation of GD in the five regions of South Korea and the other region of native accessions of *Perilla frutescens* var. *frutescens* using 20 *Perilla* SSR markers will provide information for expanding our understanding of regional genetic variation in native accessions of *Perilla frutescens* var. *frutescens* in South Korea. It is also necessary to collect farmers’ lines or landraces to prevent genetic erosion of native accessions of *Perilla frutescens* var. *frutescens* in South Korea. Therefore, the information provided here will help to characterize the introduction of new plants into the germplasm collection for conserving the native accessions of *Perilla frutescens* var. *frutescens*. Additionally, it will help in selecting useful genetic resources for the development of leafy-vegetable and seed-oil varieties of *Perilla frutescens* var. *frutescens* for the RDA-Genebank of South Korea.

In addition, information on the GD and geographic variation of the collected genetic resources of native *Perilla frutescens* var. *frutescens* is very important for efficient management and use of those collected genetic resources at the RDA-Genebank of South Korea [14]. In this study, to understand the GD and relationships, and population structure of the 200 accessions of native *Perilla frutescens* var. *frutescens* from five regions of South Korea and another region, we used the following two statistical methods: a model-based approach with STRUCTURE software and UPGMA dendrogram with NTSYS-pc V2.1. In particular, phylogenetic relationship analysis can be used as an important method for distinguishing the genetic resources of crop species to select useful genetic resources for the development of cultivars. In our study, the STRUCTURE results revealed that the 200 accessions of native *Perilla frutescens* var. *frutescens* could be divided into three main groups and an admixed group at *K* = 3 (Figure 2 and Figure 3). At *K* = 3, Group I contained 47 *Perilla* accessions from the five regions of South Korea and another region. Group II comprised 37 *Perilla* accessions from the G3 and G5 regions of South Korea and G6 region. Group III included 86 *Perilla* accessions from the five regions of South Korea and another region. The admixed group included 28 *Perilla* accessions from the G2, G3, G4, and G5 regions of South Korea and the G6 region. In addition, the UPGMA dendrogram results showed that the 200 accessions of native *Perilla frutescens* var. *frutescens* were divided into three main groups and an outgroup with 42% genetic similarity (Figure 4). According to the results of the UPGMA and STRUCTURE analysis, there was no clear geographic location among the 200 accessions of native *Perilla frutescens* var. *frutescens* from the G1, G2, G3, G4, and G5 regions in South Korea and the G6 region (Figure 3 and Figure 4). In addition, as shown in Figure 1 and Figure 2, the clustering patterns were not clearly distinguished among the native accessions of *Perilla frutescens* var. *frutescens* from the five regions of South Korea and the other region. Additionally, the *Perilla* SSR marker-based clustering of native accessions of *Perilla frutescens* var. *frutescens* showed no similarity to the dendrogram topologies based on the morphological index. These results indicate that, in South Korea, landrace seeds of native *Perilla frutescens* var. *frutescens* may be frequently exchanged by farmers among the five regions in South Korea through various routes over a long period of time, as previously reported by Lee et al. (2002) [5], Sa et al. (2013) [9] and Oh et al. (2020) [14]. The 45 *Perilla* accessions of the other region were divided into two groups as follows: some accessions belonged to the same group as the accessions collected in the five regions of South Korea, and the rest of the accessions were included in the outgroup (Figure 4). There is no clear genetic resource collection information for these accessions of the other region, but these accessions of the outgroup can probably be considered foreign-introduced resources. Therefore, the results of the UPGMA and STRUCTURE analysis and the comparison between *Perilla* SSR markers and *Perilla* morphological distance using the Mantel test are expected to be helpful in understanding seed geographical spread of native accessions of *Perilla frutescens* var. *frutescens* in accordance with geographical distribution in South Korea.

In this study, genetic variation analysis was performed using morphological characteristics and *Perilla* SSR markers for native accessions of *Perilla frutescens* var. *frutescens* collected in South Korea and another region. In future, the results of this study are expected to provide interesting information for the conservation of these genetic resources and the selection of useful resources for the development of varieties for seeds and leafy vegetables from *Perilla frutescens* var. *frutescens* in South Korea.

## 4. Materials and Methods

### 4.1. Plant Materials

For this study, 200 accessions of native *Perilla frutescens* var. *frutescens* collected in South Korea and others (unknown accessions) were obtained from the RDA-Genebank of the Republic of Korea (http://genebank.rda.go.kr/, accessed on 1 December 2019). The accession numbers and geographic information for 200 native *Perilla* accessions are shown in Appendix A. In addition, Appendix A shows the collection areas of accessions of native *Perilla frutescens* var. *frutescens* collected from South Korea, namely Gyeonggi-do (G1, 19 accessions), Gangwon-do (G2, 13 accessions), Chungcheong-do (G3, 20 accessions), Jeolla-do (G4, 36 accessions), and Gyeongsang-do (G5, 67 accessions). Meanwhile, 45 *Perilla* accessions were classified into the other region (G6) because there was no information on where they were collected (Appendix A). These accessions of native *Perilla frutescens* var. *frutescens* were selected to provide basic information for building core groups within the accessions of *Perilla* germplasm, and also for use as breeding materials for the development of leafy vegetables or seed cultivars of *Perilla* crops from the Korean RDA-Genebank.

### 4.2. Morphological Characteristics Analyzed

To assess the morphological characteristics of the 200 accessions of native *Perilla frutescens* var. *frutescens*, seven individuals of each *Perilla* accession were grown in a field at the experimental farm of Kangwon National University, Chuncheon, Gangwon-do. Approximately 20 seeds of each *Perilla* accession were sown in a nursery bed in early May and kept in a glass house for a month. Seven seedlings of each accession were then transplanted into the field in early June 2020. We examined eight qualitative and three quantitative characteristics, as shown in Table 4, at the appropriate growth stages; these characteristics were selected based on a previous report by Lee and Ohnishi (2001) [2]. Measurements of the three quantitative characteristics, namely leaf width (QN1), leaf length (QN2), flowering time (QN3), and observation of the eight qualitative characteristics, namely color of leaf surface (QL1), color of reverse side leaf (QL2), stem color (QL3), seed color (QL4), leaf shape (QL5), degree of pubescence (QL6), seed hardness (QL7), and plant fragrance (QL8), were made on five individuals for each accession based on a previous report by Lee and Ohnishi (2001) [2]. However, 17 *Perilla* accessions in our study were not used for measurement because of plant growth failure in the field. Therefore, we conducted a morphological characteristics survey of only 183 accessions out of the 200 accessions of native *Perilla frutescens* var. *frutescens* (Appendix A).

### 4.3. SSR Analysis and DNA Electrophoresis

For SSR analysis, the total DNA of 200 *Perilla* accessions was extracted from the leaf tissue of an individual plant of each *Perilla* accession using the Plant DNAzol Reagent protocols (GibcoBRL Inc., Grand Island, NY, USA). In a preliminary test with over 100 *Perilla* SSR primer sets developed by many researchers [18,19,23,39,40], we selected 20 *Perilla* SSR primer sets that showed a clear banding pattern and high allele band amplification in *Perilla* accessions (Appendix A). The SSR amplification method for the *Perilla* crop has been described in a previous study by Kim et al. (2021) [23]. After polymerase chain reaction (PCR) amplification using *Perilla* SSR primer sets, DNA electrophoresis analysis was performed with a QIAxcel advanced system (QIAGEN Co., Hilden, Germany) using the protocol described in the QIAxcel DNA Handbook. The amplification products were electrophoresed in the QIAxcel advanced electrophoresis system, and amplification product separation was performed over 15 min. Gel images were obtained as the amplification results, and the quantification analysis was performed with QIAxcel software. The amplification results were displayed as gel images and electropherograms taken from the QIAxcel advanced system software based on a previous report by Park et al. (2021) [13].

### 4.4. Data Analysis

DNA fragments amplified for each *Perilla* SSR primer set were scored as present (1) or absent (0). Power Marker version 3.25 [41] was applied to obtain information on the number of alleles, allele frequency, MAF, GD, and PIC. GS was calculated for each pair of accessions using the Dice similarity index [42]. To illustrate the genetic relationships of the total accessions, a similarity matrix was used to construct UPGMA dendrogram by the application of SAHN-Clustering from NTSYS-pc V2.1 (Exeter Software, Setauket, NY, USA) [43]. Population structure was investigated for the 200 accessions of native *Perilla frutescens* var. *frutescens* using STRUCTURE 2.2 software [44]. Five independent runs with *K* values ranging from one to ten were performed with 100,000 cycles for both burn-in and run length. The delta *K* statistic, based on the rate of change in the log probability of data between *K* values [45], was calculated with STRUCTURE HARVESTER (http://taylor0.biology.ucla.edu/structHarvester/, accessed on 1 November 2020) based on the STRUCTURE results. Cluster analysis for morphological characteristics was performed using NTSYS-pc V2.1 [43]. The correspondence between the morphology and the SSR loci-based similarity coefficient matrices was tested on the basis of correlation analysis, and the Mantel (1967) [46] matrix correspondence test was carried out using the MXCOMP procedure in NTSYS-pc V2.1 [43]. SPSS software (IBM Corporation, Armonk, N.Y., USA) was used to perform correlation analysis for three quantitative and eight qualitative characteristics of the 183 accessions of native *Perilla frutescens* var. *frutescens*.

## 5. Conclusions

In this study, we evaluated the genetic variation and population structure of 200 accessions of native *Perilla frutescens* var. *frutescens* collected from five regions of South Korea and another region using *Perilla* SSR markers and morphological characteristics. Among the morphological traits investigated in our study, the accessions of native *Perilla frutescens* var. *frutescens* showed high frequency in the following traits: green for leaf and stem color, brown and dark brown for seed color, lanceolate and heart for leaf shape, slightly pubescent and normal pubescent for degree of pubescence, soft seed for seed hardness, the specific plant fragrance of native *Perilla frutescens* var. *frutescens* for plant fragrance, a size of 8.1–10.0 cm for leaf width, a size of 12.1–14.0 cm for leaf length, and intermediate maturing type for flowering time. These results indicate that the accessions of native *Perilla frutescens* var. *frutescens* exhibiting highest frequency characteristics for each morphological trait are related to their use as leaves or seeds in South Korea. In addition, the results of clustering patterns based on the morphological traits were not clearly distinguished among the accessions of native *Perilla frutescens* var. *frutescens* from the five regions of South Korea and another region. Additionally, most *Perilla* accessions in each group did not show any particular morphological differences among the five regions of South Korea and the other region. For analysis of *Perilla* SSR markers, the accessions of the Jeolla-do (G4) and Gyeongsang-do (G5) regions showed a comparatively high GD value compared with those from other regions in South Korea. In the population structure and UPGMA analysis, there was no clear geographic structure among the 200 accessions of native *Perilla frutescens* var. *frutescens* from the five regions of South Korea and another region. These results indicate that, in South Korea, landrace seeds of native *Perilla frutescens* var. *frutescens* may be frequently exchanged by farmers among the five regions in South Korea through various routes. The results of this study are expected to provide useful information for the conservation of these genetic resources and the selection of useful resources for the development of varieties for seeds and leafy vegetables of cultivated *Perilla frutescens* var. *frutescens* in South Korea.

## Figures and Tables

**Figure 1 plants-10-01764-f001:**
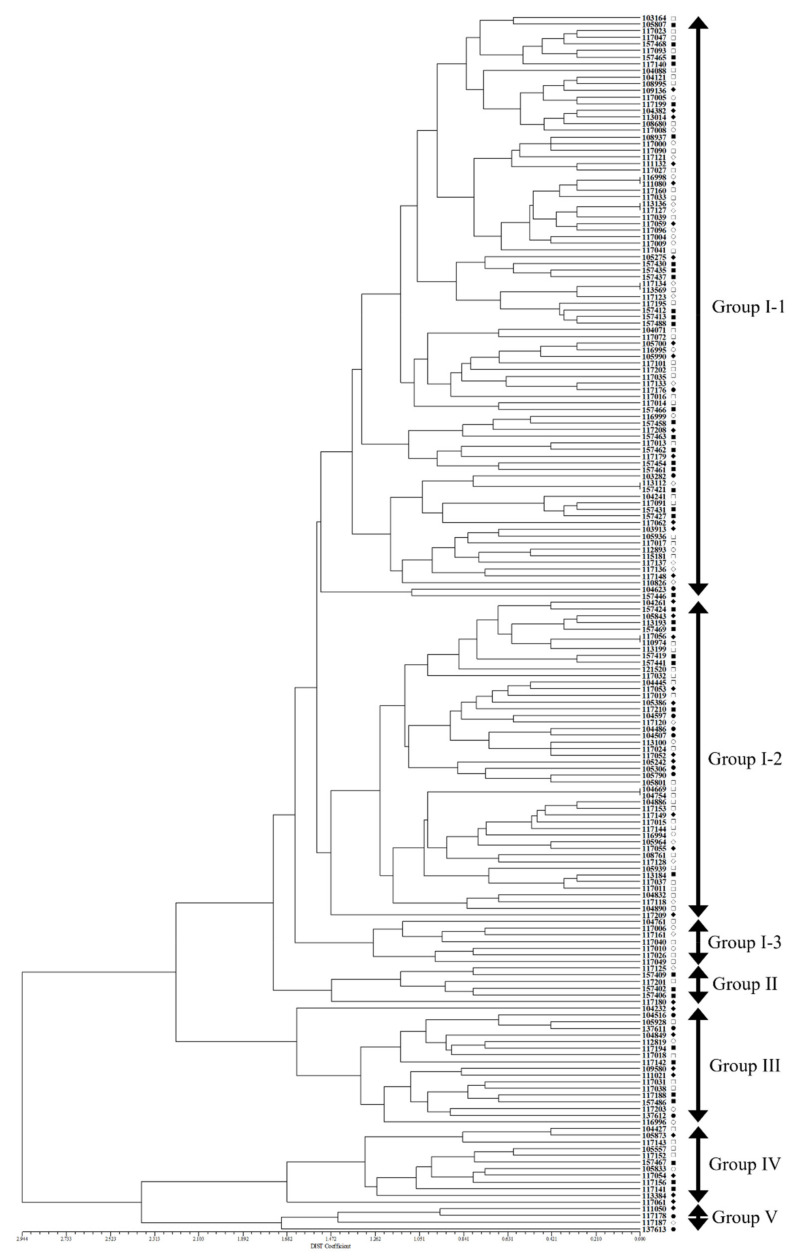
Dendrogram of the 183 accessions of native *Perilla frutescens var. frutescens* based on 11 morphological traits. The major clusters are marked on the right side of the dendrogram. The scale at the bottom is DIST coefficient of dissimilarity. ○: Gyeonggi-do, ●: Gangwon-do, ◇: Chungcheong-do, ◆: Jeolla-do, □: Gyongsang-do, ■: others.

**Figure 2 plants-10-01764-f002:**
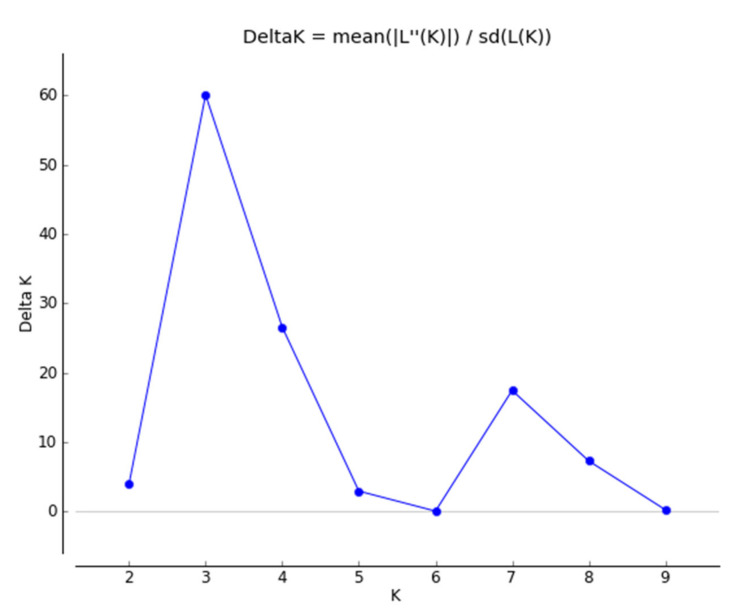
Magnitude of ΔK as a function of K. The peak value of ΔK was at *K* = 3, suggesting the existence of three main groups and an admixed group in the 200 accessions of the native *Perilla frutescens* var. *frutescens* of *Perilla* crop.

**Figure 3 plants-10-01764-f003:**
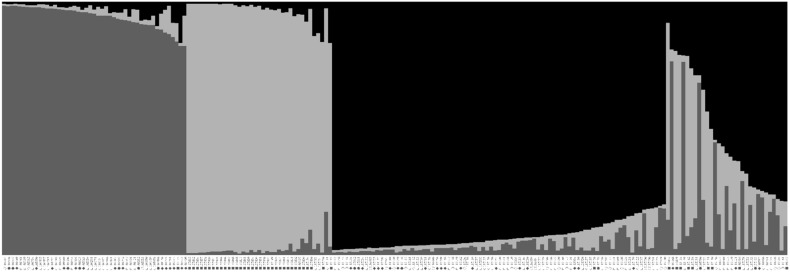
Population structure pattern for the highest *ΔK* value (*K* = 3) of 200 accessions of native *Perilla frutescens* var. *frutescens* of *Perilla* crop. ○: Gyeonggi-do, ●: Gangwon-do, ◊: Chungcheong-do, ◆: Jeolla-do, □: Gyongsang-do, ■: other region.

**Figure 4 plants-10-01764-f004:**
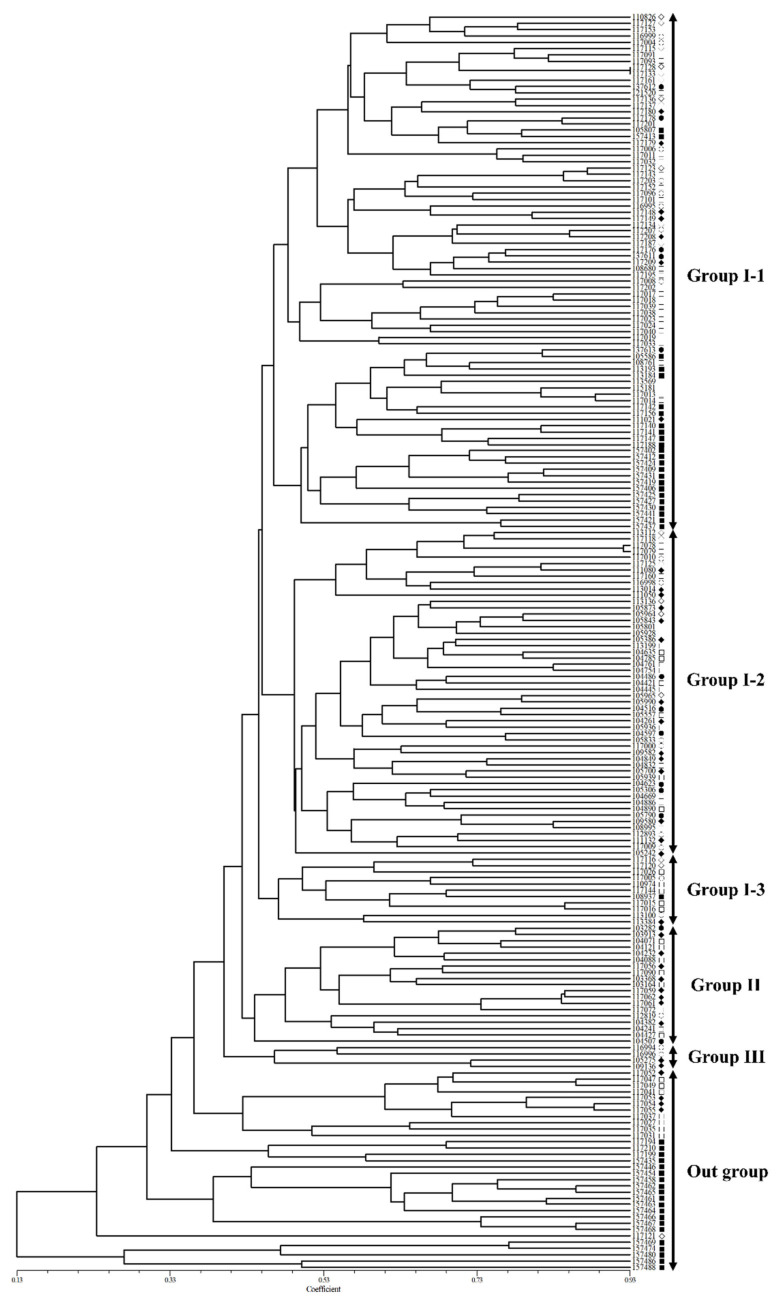
UPGMA dendrogram of 200 accessions of the native *Perilla frutescens* var. *frutescens* based on 20 SSR markers. The solid line represents the 11 major groups. ○: Gyeonggi-do, ●: Gangwon-do, ◊: Chungcheong-do, ◆: Jeolla-do, □: Gyongsang-do, ■: other region.

**Figure 5 plants-10-01764-f005:**
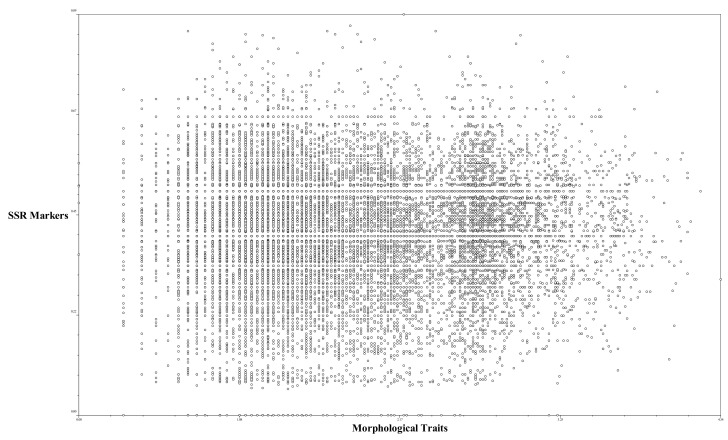
Comparison between SSR markers and morphological distance using the Mantel test. Correlation coefficient is 0.01.

**Table 1 plants-10-01764-t001:** Correlation coefficients among 11 qualitative and quantitative characters in 183 accessions of native *Perilla frutescens* var. *frutescens* of *Perilla* crop.

	QL2	QL3	QL4	QL5	QL6	QL7	QL8	QN1	QN2	QN3
QL1	−0.133	1.000 **	0.011	0.109	−0.088	0.051	−0.012	0.139	0.104	−0.049
QL2		−0.133	0.095	−0.030	−0.120	−0.045	−0.084	0.013	−0.071	0.011
QL3			0.011	0.109	−0.088	0.051	−0.012	0.139	0.104	−0.049
QL4				0.191 **	−0.018	0.050	0.026	0.239 **	0.146 *	−0.032
QL5					0.073	−0.047	−0.057	0.031	0.032	−0.153 *
QL6						0.155 *	0.151 *	0.050	0.036	0.166 *
QL7							0.070	−0.019	−0.065	−0.087
QL8								−0.052	−0.041	0.043
QN1									0.534 **	0.172 *
QN2										0.162 *

** Significance at *p* < 0.01, * Significance at *p* < 0.05.

**Table 2 plants-10-01764-t002:** Characteristics of the 20 *Perilla* SSR loci including allele size, number of allele, MAF, GD, PIC among 200 accessions of native *Perilla frutescens* var. *frutescens* of *Perilla* crop.

SSR Loci	Allele Size	No. of Allele	MAF	GD	PIC
KNUPF1	176–182	5	0.835	0.290	0.272
KNUPF2	156–166	9	0.385	0.726	0.680
KNUPF4	158–168	7	0.370	0.708	0.654
KNUPF5	177–183	6	0.470	0.693	0.651
KNUPF10	195–220	13	0.565	0.648	0.626
KNUPF16	141–190	13	0.225	0.828	0.805
KNUPF23	204–214	4	0.565	0.592	0.532
KNUPF25	179–190	5	0.485	0.629	0.561
KNUPF26	153–264	3	0.570	0.571	0.501
KNUPF28	174–192	6	0.660	0.522	0.484
KNUPF33	186–213	11	0.335	0.788	0.760
KNUPF36	170–180	6	0.390	0.749	0.712
KNUPF37	256–259	4	0.490	0.641	0.577
KNUPF39	196–223	9	0.410	0.711	0.662
KNUPF43	127–155	10	0.255	0.791	0.759
KNUPF55	233–253	11	0.520	0.681	0.653
KNUPF59	152–170	4	0.580	0.525	0.431
KNUPF71	182–185	4	0.435	0.645	0.572
KNUPF74	190–198	4	0.445	0.597	0.512
KNUPF77	166–174	3	0.405	0.651	0.576
Total		137			
Mean		6.85	0.470	0.649	0.599

MAF: major allele frequency, GD: genetic diversity, PIC: polymorphic information content.

**Table 3 plants-10-01764-t003:** Estimates of MAF, allele number, genetic diversity and PIC of 20 *Perilla* SSR loci collected in five regions from South Korea and other region.

Marker	No. of Alleles						Major Allele Frequency						Genetic Diversity						PIC					
	G1	G2	G3	G4	G5	G6	G1	G2	G3	G4	G5	G6	G1	G2	G3	G4	G5	G6	G1	G2	G3	G4	G5	G6
KNUPF1	2	2	2	4	3	4	0.947	0.923	0.900	0.889	0.851	0.667	0.100	0.142	0.180	0.205	0.265	0.491	0.095	0.132	0.164	0.198	0.249	0.433
KNUPF2	6	3	6	6	5	4	0.632	0.462	0.450	0.500	0.463	0.800	0.565	0.639	0.705	0.668	0.644	0.343	0.536	0.566	0.663	0.624	0.579	0.320
KNUPF4	5	4	5	4	6	6	0.474	0.538	0.450	0.472	0.418	0.511	0.676	0.627	0.630	0.597	0.664	0.674	0.627	0.576	0.560	0.515	0.600	0.637
KNUPF5	5	4	3	4	6	5	0.737	0.692	0.500	0.694	0.507	0.444	0.438	0.485	0.605	0.485	0.661	0.688	0.416	0.451	0.527	0.452	0.616	0.638
KNUPF10	5	4	6	6	10	6	0.368	0.615	0.250	0.528	0.672	0.667	0.726	0.568	0.800	0.651	0.534	0.528	0.680	0.526	0.770	0.610	0.520	0.504
KNUPF16	5	5	5	8	11	7	0.316	0.385	0.400	0.250	0.284	0.689	0.765	0.722	0.715	0.823	0.811	0.505	0.726	0.676	0.668	0.799	0.785	0.485
KNUPF23	4	2	3	4	3	4	0.579	0.538	0.650	0.389	0.552	0.711	0.560	0.497	0.515	0.677	0.594	0.454	0.488	0.374	0.460	0.611	0.528	0.413
KNUPF25	4	3	3	4	4	4	0.526	0.385	0.700	0.444	0.448	0.667	0.632	0.651	0.445	0.650	0.619	0.516	0.578	0.576	0.381	0.583	0.540	0.479
KNUPF26	3	3	3	3	3	3	0.474	0.462	0.750	0.444	0.537	0.711	0.637	0.615	0.395	0.634	0.591	0.440	0.565	0.535	0.347	0.558	0.517	0.386
KNUPF28	4	4	3	4	6	5	0.684	0.462	0.800	0.611	0.642	0.711	0.493	0.627	0.335	0.560	0.532	0.443	0.456	0.556	0.303	0.508	0.486	0.392
KNUPF33	6	4	7	7	7	7	0.421	0.538	0.250	0.472	0.343	0.356	0.704	0.627	0.820	0.715	0.778	0.784	0.657	0.576	0.795	0.684	0.747	0.757
KNUPF36	4	4	4	5	6	5	0.526	0.462	0.550	0.333	0.358	0.489	0.626	0.686	0.625	0.776	0.761	0.683	0.568	0.637	0.578	0.742	0.725	0.643
KNUPF37	3	3	3	4	4	4	0.474	0.462	0.800	0.500	0.582	0.578	0.632	0.639	0.335	0.619	0.578	0.591	0.558	0.566	0.303	0.549	0.519	0.537
KNUPF39	4	4	3	5	6	6	0.474	0.615	0.550	0.389	0.388	0.378	0.654	0.556	0.535	0.715	0.719	0.721	0.594	0.506	0.436	0.664	0.672	0.672
KNUPF43	6	5	5	5	8	5	0.368	0.308	0.400	0.472	0.313	0.667	0.770	0.746	0.665	0.705	0.794	0.493	0.738	0.702	0.604	0.670	0.764	0.437
KNUPF55	7	6	6	6	6	6	0.263	0.308	0.450	0.389	0.627	0.689	0.798	0.781	0.710	0.764	0.566	0.502	0.769	0.748	0.670	0.733	0.533	0.479
KNUPF59	3	3	3	3	4	2	0.632	0.538	0.650	0.694	0.537	0.511	0.521	0.556	0.485	0.440	0.558	0.500	0.455	0.465	0.406	0.365	0.468	0.375
KNUPF71	3	2	3	4	2	3	0.737	0.538	0.550	0.472	0.507	0.822	0.410	0.497	0.535	0.576	0.500	0.307	0.359	0.374	0.436	0.484	0.375	0.284
KNUPF74	2	3	3	3	3	4	0.632	0.538	0.600	0.444	0.507	0.489	0.465	0.556	0.560	0.610	0.589	0.595	0.357	0.465	0.499	0.527	0.506	0.513
KNUPF77	3	3	3	3	3	3	0.474	0.385	0.450	0.556	0.463	0.467	0.615	0.663	0.615	0.586	0.638	0.631	0.536	0.589	0.534	0.517	0.564	0.556
Averge	4	4	4	5	5	5	0.537	0.508	0.555	0.497	0.500	0.601	0.589	0.594	0.561	0.623	0.620	0.544	0.538	0.530	0.505	0.570	0.565	0.497

G1: Gyeonggi-do (19 accessions), G2: Gangwon-do (13 accessions), G3: Chungcheong-do (20 accessions), G4: Jeolla-do (36 accessions), G5: Gyongsang-do (67 accessions), G6: other region (45 accessions).

**Table 4 plants-10-01764-t004:** Characters used in the morphological analysis of the 183 accessions of native *Perilla*
*frutescens* var. *frutescens* of *Perilla* crop.

Morphological Character		When/how Measured	Category or Unit
QL1	Color of leaf surface	at flowering stage	1-light green, 2-green, 3-deep green
QL2	Color of reverse side leaf	at flowering stage	1-light green, 2-green, 3-deep green
QL3	Stem color	at flowering stage	1-light green, 2-green, 3-deep green
QL4	Seed color	after harvest	1-white, 2-gray, 3-brown, 4-dark brown
QL5	Leaf shape	at flowering stage	1-lanceolate, 2-heart shape, 3-oblong
QL6	Degree of pubescence	at flowering stage	1-slightly pubescent, 2-normal pubescent, 3-heavily pubescent
QL7	Seed hardness	after harvest	1-soft, 2-hard
QL8	Plant fragrance	at flowering stage	1-plant fragrance of var. *frutescens*, 2-other plant fragrance
QN1	Leaf width	at flowering stage	1–8.0 cm ≤, 2–8.1~9.0 cm, 3–9.1~10.0 cm, 4–10.1~11.0 cm, 5–11.1~12.0 cm, 6–12.1~13.0 cm, 7–13.1 cm >
QN2	Leaf length	at flowering stage	1–10.0 cm ≤, 2–10.1~11.0 cm, 3–11.1~12.0 cm, 4–12.1~13.0 cm, 5–13.1~14.0 cm, 6–14.1~15.0 cm, 7–15.1 cm >
QN3	Flowering time	at flowering stage (the day of more than 50% flowering per plant)	1-early maturing type (flowering days before 15 August), 2-intermediate maturing type (flowering days from 15 August to 5 September), 3-late maturing type (flowering days after 6 September)

## Data Availability

Data is contained within the article or Appendix A.

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
