# Peer review of "Genetic Variation of Native Perilla Germplasms Collected from South Korea Using Simple Sequence Repeat (SSR) Markers and Morphological Characteristics"

_plants, 2021, doi:10.3390/plants10091764_

Round 1

Reviewer 1 Report

Using 200 accessions of native Perilla germplasms collected from South Korea, the authors studied the genetic variation of morphological parameters and SSR markers.

I also appreciate the authors efforts for using a wide set of accessions.

I have many comments about the manuscript.

Minor comments

-In introduction must be clarified the number of the chromosome in Perilla.

-In results: Table 3 must be landscape format.

Major comments

In materials and methods

- Type experimental design and statistical analysis of morphological data don't exist.

- 20 SSRs markers were using only, they are very small. And how many chromosomes are covered?

- How principal component analysis (PCA) was performed using the Microsoft Excel Statistical Analysis System Program?

In results:

-The method of calculation principal component is not correct, because the sum of squares any component equals 1.

- Principal component analysis is used to reduce the number of variables and overcome the problems of multicollinearity and /or as a step before making analyses multiple regression, path analysis, and discriminant analysis. What is the purpose of using PCA in that study?

- Cluster analysis should be calculated for morphological parameters too, and evaluation of genetic distance with SSRs data by the Mantel test.

Author Response

Reviewer #1:

Using 200 accessions of native Perilla germplasms collected from South Korea, the authors studied the genetic variation of morphological parameters and SSR markers. I also appreciate the authors efforts for using a wide set of accessions. I have many comments about the manuscript.

Minor comments

-In introduction must be clarified the number of the chromosome in Perilla.

-> According to the reviewer’s comments, in the new version manuscript, we showed the number of the chromosome in Perilla as follows; Even though the two cultivated types of Perilla crop have different morphological features, they have same number of chromosomes of tetraploids (2n=40) [2, 4, 6]. See L. 46 - 48, in new version manuscript.

-In results: Table 3 must be landscape format.

-> Yes, Table 3 is landscape format.

Major comments

In materials and methods

- Type experimental design and statistical analysis of morphological data don't exist.

-> According to the reviewer’s comments, in the new version manuscript, we explained about “Type experimental design and statistical analysis of morphological data” as follow: “Measurements of the three quantitative characteristics, ------based on the previous report by Lee and Ohnishi (2001) [2].

In addition, we presented the survey method for each morphological trait in more detail in Table 4.

See L. 430 - 431, and Table 4 (L. 435-437), in new version manuscript.

- 20 SSRs markers were using only, they are very small. And how many chromosomes are covered?

-> Unfortunately, Perilla crop is one of the minor crop in the world, so currently only 120 SSR markers developed by our research team and several Korean researchers. In addition, unfortunately, there is still no information about the function and chromosomal location of the SSR markers selected in our study. As described on L. 441- 444, in new version manuscript, in our study, 20 SSR primer sets were selected in a preliminary test of 100 SSR primer sets to identify efficient SSR primer sets showing high allele band amplification and a clear banding pattern. Therefore, these SSR markers could be useful for genetic diversity of native Perilla germplasms collected from South Korea

- How principal component analysis (PCA) was performed using the Microsoft Excel Statistical Analysis System Program?

-> In our study, we used the Microsoft Excel Statistical Analysis System Program to basic data input file for NTSYS-pc V2.1 Program execution, and to graphically represent based on the PCA analysis results obtained by the NTSYS-pc V2.1 Program, as shown in Figure 1.

In results:

-The method of calculation principal component is not correct, because the sum of squares any component equals 1.

-> In this study, we only presented the cumulative variances of first and second principal components from the results of PCA analysis. In addition, we think the main reason for the low cumulative variances of first and second principal components, because the morphological traits used in the analysis did not show much morphological variation among 183 accessions of cultivated Perilla germplasm in South Korea.

- Principal component analysis is used to reduce the number of variables and overcome the problems of multicollinearity and /or as a step before making analyses multiple regression, path analysis, and discriminant analysis. What is the purpose of using PCA in that study?

-> Principal component analysis was commonly used because it was a useful analytical method for measuring inter- and intra-individual morphological variability in many plant populations.

In particular, PCA analysis is a useful analytical method that enables comparative analysis of quantitative and qualitative traits between and within in the plant populations. Therefore, in our study, we used PCA analysis on 183 accessions of cultivated Perilla germplasm in South Korea to understand morphological variations.

- Cluster analysis should be calculated for morphological parameters too, and evaluation of genetic distance with SSRs data by the Mantel test.

-> In this study, we tried to understand the morphological variation and genetic diversity of Korean native Perilla germplasm using morphological traits and SSR markers.

Although the reviewer suggested good opinions on the genetic diversity evaluation method using both morphological and SSR data, in this study, cluster analysis was measured using only SSR markers for more accurate genetic diversity analysis. This is because morphological traits are more affected by environmental influences than molecular markers.

In addition, a method for assessing the genetic distance using SSR data by applying SAHN-clustering from NTSYS-pc V2.1, which has already been established by many researchers.

Reviewer 2 Report

The manuscript entitled: ” Genetic variation of native Perilla germplasms collected from  South Korea using SSR markers and morphological characteristics” presents an interesting study regarding the diversity of Perilla germplasm, providing useful information for the conservation of these resources. I have some comments for the authors to improve the manuscript:

  1. Don’t use abbreviations in the title, eg. Explain SSR
  2. In the abstract explain SSR at the first use in the text – line 11 not at the second use – line 16
  3. Define all the abbreviations used in the manuscript at the first used, eg. MAF, PIC value,
  4. In the material and methods for software used add the manufacturer and country
  5. Strengths and limitations of the study should be added in the discussion section

Author Response

I have some comments for the authors to improve the manuscript:

1.Don’t use abbreviations in the title, eg. Explain SSR

-> In revised version manuscript, we changed “SSR” to “simple sequence repeat (SSR)” in the Title. See L. 2, in new version manuscript.

2.In the abstract explain SSR at the first use in the text – line 11 not at the second use – line 16

-> In revised version manuscript, we explained “SSR” to “simple sequence repeat (SSR)” at the first use in the abstract. See L. 11, in new version manuscript.

3.Define all the abbreviations used in the manuscript at the first used, eg. MAF, PIC value,

-> In revised version manuscript, we defined all the abbreviations (eg. MAF, PIC value) used in the manuscript at the first used. See L. 179-180, in new version manuscript.

4.In the material and methods for software used add the manufacturer and country

-> In revised version manuscript, we added the manufacturer and country about the software. See L. 489, L. 500, in new version manuscript.

5.Strengths and limitations of the study should be added in the discussion section

-> We already explained strengths and limitations of this study in the Conclusion. In revised version manuscript, we modified some of the texts in the Conclusion. See L. 518-522, in new version manuscript.

Reviewer 3 Report

non

Author Response

Thank you for your comments

Reviewer 4 Report

plants-1328655: Genetic variation of native Perilla germplasms collected from South Korea using SSR markers and morphological characteristics

Comments:

Line 87-88: Statement is not clear and meaningful. Write the objective of the study clearly.

Table 1: Are the colour of leaf, seed color, stem colour, leaf length, leaf width, flowering time etc. correlated to each other? If these traits are not correlated, a PCA is not valid. A greater coefficient of 0.909 for stem colour and colour of leaf surface in PC1 indicated these two variables dominated across PC1.

In section 4.4 (line 4.39) the authors stated that a correlation analysis was conducted using SPSS software. Did you present the output anywhere in the manuscript or as supplementary materials?

Present all data as supplementary files so that the PCA analysis can be reviewed.

Figure 1: PC1 and PC2 collectively explained only 35% data variation. Thus 65% data variation remains unexplained. Any conclusion drawn based on only 35% data variation is insufficient.

Clearly explain the basis of three groups obtained across the PC1, in discussion.

In general, much logical improvement is needed in discussion.

What was the actual number of accessions studied? In Table S1 the number is 183. In other places of the manuscript the number is 200 (see lines 12, 143 etc). Why different number of accessions were used in between morphological and SSR-based study?

Figure S2: How did you make these four groups? Describe in methodology.

Figure S3: Define all traits in Supplementary figures 1 and 3.

References should be written following the MDPI style and consistently (see lines 535, 584).  

Author Response

Reviewer #3:

Comments:

- Line 87-88: Statement is not clear and meaningful. Write the objective of the study clearly.

->In new version manuscript, according to the reviewer’s comments, we added the objective of the study as follows: The results of this study are expected to provide useful information to understand the genetic variation of native Perilla germplasm collected from South Korea.

See L. 92 - 94, in new version manuscript.

Table 1: Are the colour of leaf, seed color, stem colour, leaf length, leaf width, flowering time etc. correlated to each other? If these traits are not correlated, a PCA is not valid. A greater coefficient of 0.909 for stem colour and colour of leaf surface in PC1 indicated these two variables dominated across PC1.

-> The traits used in this study to investigate the morphological characteristics are traits related to leaf quality and seed quality of Perilla cultivars mainly used in Korea. Therefore, these traits are thought to provide useful information for understanding the cultivar differentiation and regional variation of native Perilla accessions that has been cultivated for a long time in Korea.

And also, PCA analysis is a useful analytical method that enables comparative analysis of quantitative and qualitative traits between and within individuals in a native Perilla accessions. Therefore, in our study, we used PCA analysis on the 183 accessions of cultivated Perilla germplasm in South Korea to understand morphological variations such as morphological characteristics related to leaf quality and seed quality of Korean Perilla cultivars.

In section 4.4 (line 4.39) the authors stated that a correlation analysis was conducted using SPSS software. Did you present the output anywhere in the manuscript or as supplementary materials?

-> In new version manuscript, we deleted the sentence “SPSS software was used to perform correlation analysis for three quantitative and eight qualitative characteristics”.

See L. 457, in new version manuscript.

Present all data as supplementary files so that the PCA analysis can be reviewed.

-> We present all data (raw data and results performed by the Microsoft Excel Statistical Analysis System Program and NTSYS-pc V2.1 Program) of the PCA analysis as supplementary file.

See supplementary file.

- Figure 1: PC1 and PC2 collectively explained only 35% data variation. Thus 65% data variation remains unexplained. Any conclusion drawn based on only 35% data variation is insufficient.

-> In this study, we only presented the cumulative variances of first and second principal components from the results of PCA analysis. In addition, we think the main reason for the low cumulative variances of first and second principal components, because the morphological traits used in the analysis did not show much morphological variation among 183 accessions of cultivated Perilla germplasm in South Korea.

- Clearly explain the basis of three groups obtained across the PC1, in discussion.

In general, much logical improvement is needed in discussion.

-> According to the reviewer’ comment, we explained the basis of three groups obtained across the PC1, in discussion as follows; In particularly, according to the results of PCA analysis, among the traits used in the analysis, QL1 and QL3, and QL2 and QL6 traits greatly contributed in the positive and negative direction on the first axis. As shown in Fig. 1, 183 accessions of cultivated P. frutescens var. frutescens were divided into three groups on the first axis. In addition, of the three groups, most accessions showing green leaf and stem color with slightly pubescent are located in the center, and most accessions showing light green leaf and stem color with normal pubescent are located in the negative side on the first axis. On the other hand, most accessions located on the positive side showed deep green leaf and stem color with heavily pubescent. Therefore, the first axis could mainly be utilized for distinction among 183 accessions of cultivated P. frutescens var. frutescens in South Korea. However, most accessions collected in South Korea did not clearly showed their collection area on the first axis. These results suggest that the morphological characteristics of cultivated P. frutescens var. frutescens used in South Korea are similarly favored by farmers in all regions.

See L. 301 - 314, in new version manuscript.

- What was the actual number of accessions studied? In Table S1 the number is 183. In other places of the manuscript the number is 200 (see lines 12, 143 etc). Why different number of accessions were used in between morphological and SSR-based study?

-> As already explained in the materials and methods, 17 of the 200 accessions used in the SSR analysis were not used for morphological analysis due to cultivation failure in the field.

See L. 431 - 433, in new version manuscript.

Figure S2: How did you make these four groups? Describe in methodology.

-> In old version manuscript, we already explained about the Figure S2 as follows; “Among the 137 alleles, 31 private alleles (23%) were only detected in one of the 200 accessions of cultivated P. frutescens var. frutescens. The percentage of rare alleles (frequency < 0.05) was 45% (62 alleles) among the 137 alleles, whereas intermediate-frequency alleles (frequency of 0.05-0.5) and abundant alleles (frequency > 0.5) represented 50% (68 alleles) and 5% (7 alleles), respectively, of the total alleles (Supplement Fig. 2).”. The Figure S2 showed the result of analysis using the statistical program of STRUCTURE 2.2 software for the 137 alleles detected in 200 Perilla accessions by 20 SSR markers. Namely it showed a segregation pattern of 137 alleles in 200 Perilla accessions.

See L. 152 - 157, in new version manuscript.

Figure S3: Define all traits in Supplementary figures 1 and 3.

-> In new version manuscript, we already explained about the Supplementary figures 1 and 3 as follows; Figure S1: Morphological features of two cultivated types of Perilla crop grown for investigation of morphological characteristics in experimental field: (a) Perilla frutescens var. frutescens and (b) P. frutescens var. crispa. Figure S3: Collection areas of 200 accessions of cultivated type of P. frutescens var. frutescens collected from five regions of South Korea and other region. For the regions of South Korea, 155 accessions were collected from Gyeonggi-do (G1, 19 accessions), Gangwon-do (G2, 13 accessions), Chungcheong-do (G3, 20 accessions), Jeolla-do (G4, 36 accessions), and Gyeongsang-do (G5, 67 accessions). However, collection areas of 45 accessions (G6 region) are not shown because there was no information on where they were collected.

See L. 497-509, in new version manuscript.

In addition, the Supplement Table 1shows the morphological variations of 183 accessions of cultivated P. frutescens var. frutescens of Perilla crop for 11 qualitative and quantitative traits in the five regions of South Korea and other region.

See Supplement Table 1.

References should be written following the MDPI style and consistently (see lines 535, 584).

-> We corrected the References according to the regulations of the MDPI style.

See L. 525, 534, 540, 550, 599, 603, in new version manuscript.

In addition, we changed Acknowledgments as follows: This study was supported by the Cooperative Research Program for Agriculture Science and Technology Development (project nos. PJ014227032019 and PJ0142272019), Rural Development Administration, Republic of Korea.

See L. 518-520, in new version manuscript.

Round 2

Reviewer 1 Report

Authors must do the following:
 - The principal component calculation correctly (it is not correct), because the sum of squares any component for eigenvectors equals 1.
-  Cluster analysis should be calculated for morphological parameters too, and evaluation of genetic distance with SSRs data by the Mantel test.

Author Response

Authors must do the following:

- The principal component calculation correctly (it is not correct), because the sum of squares any component for eigenvectors equals 1.

-> In revised version manuscript, we deleted about the results of the PCA analysis. See the new version manuscript.

- Cluster analysis should be calculated for morphological parameters too, and evaluation of genetic distance with SSRs data by the Mantel test.

-> In revised version manuscript, we added Cluster analysis by calculated for morphological parameters, and correspondence between SSR markers and morphological distance using the Mantel test.

See L. 126-154, L. 270-280, L. 411-415, L. 424-425, L. 496-502, L. 643, and new Figures 2 and 5, in new version manuscript.

Reviewer 2 Report

The authors addressed all my comments so now the manuscript is ready for acceptance. 

Reviewer 4 Report

Thanks to the authors for providing a new version of the manuscript. The authors addressed some comments however; some major comments were not addressed at the scientifically acceptable level.

  1. The traits under the study were not correlated (see attached file). Hence, a PCA analysis is not acceptable.
  2. Morphological data show that there were no replications.

      3. Rather than incorporating the output of 'SPSS based correlation analysis' the authors deleted the statement.

4. The authors failed to revise the legend of the supplementary figures.

Author Response

1.The traits under the study were not correlated (see attached file). Hence, a PCA analysis is not acceptable.

-> In revised version manuscript, we deleted about the results of the PCA analysis. See the new version manuscript.

2.Morphological data show that there were no replications.

-> As shown in Supplement Table 1, and Table 4, the phenotypic characteristics survey in this study evaluated mainly the frequency of 183 accessions of cultivated var. frutescens for qualitative and quantitative traits. See Supplement Table 1, and Table 4, in the new version manuscript.

3. Rather than incorporating the output of 'SPSS based correlation analysis' the authors deleted the statement.

-> In revised version manuscript, we added about the 'SPSS based correlation analysis'. See L. 126-171, L. 320-327, L. 500-502, and new Table 1, in the new version manuscript.

4. The authors failed to revise the legend of the supplementary figures.

-> I do not clearly understand the reviewer’s comment. In revised version manuscript, we modified some of the texts of Supplementary figures 1 and 3. See L. 533-539, in the new version manuscript.